



# Influence of Meteoric Smoke Particles on the Incoherent Scatter Measured with EISCAT VHF

Tinna L. Gunnarsdottir[1], Ingrid Mann[1], Wuhu Feng[2], Devin R. Huyghebaert[1], Ingemar Haeggstroem[3], Yasunobu Ogawa[4], Norihito Saito[5], Satonori Nozawa[5], and Takuya D. Kawahara[5]

[1]Department of Physics and Technology, UiT The Arctic University of Norway, Tromsø, Norway
[3]EISCAT Scientific Association, Kiruna, Sweden
[4]National Institute of Polar Research, JAPAN
[5]Institute for Space-Earth Environmental Research; Nagoya University, Japan
[2]National Centre for Atmospheric Science, University of Leeds, United Kingdom

**Correspondence:** Ingrid Mann (ingrid.b.mann@uit.no)

**Abstract.** Meteoric ablation in the Earth's atmosphere produces particles of nanometer-size and larger. These particles can become charged and influence the charge balance in the D-region (60-90 km) and the incoherent scatter observed with radar from there. Radar studies have shown that if enough dust particles are charged, they can influence the received radar spectrum below 100 km, provided the electron density is sufficiently high ($>10^9$ $m^3$). Here, we study an observation made with the EISCAT VHF radar on 9 January 2014 during strong particle precipitation so that incoherent scatter was observed down to almost 60 km altitude. We found that the measured spectra were too narrow in comparison to the calculated spectra. Adjusting the collision frequency provided a better fit in the frequency range $\pm$ 10-30 Hz. However, this did not lead to the best fit in all cases, especially not for the central part of the spectra in the narrow frequency range of $\pm$ 10 Hz. By including a negatively charged dust component, we obtained a better fit for spectra observed at altitudes 75-85 km, indicating that dust influences the incoherent scatter spectrum at D-region altitudes. The observations at lower altitudes were limited by the small amount of free electrons, and observations at higher altitudes were limited by the height resolution of the observation. Inferred dust number densities range from a few particles up to $10^4 cm^{-3}$ and average sizes range from approximately 0.6 to 1 nm. We find an acceptable agreement with the dust profiles calculated with the WACCM-CARMA model. However, these do not include charging, which is also based on models.

## 1 Introduction

Cosmic dust material enters Earth's atmosphere each day, globally around $25.0 \pm 7.0$ tonnes/day as recently suggested (Hervig et al., 2021). Much of this material ablates in the altitude region of 70-110 km (Plane, 2012). This meteoric material recondenses to form nanometer-sized solid dust called meteoric smoke particles (Hunten et al., 1980; Rosinski and Snow, 1961). These particles influence the charge balance in the D-region ionosphere (Baumann et al., 2015); and they possibly facilitate the nucleation of ice particles in the cold summer mesopause (Rapp and Lübken, 2004). The distribution of meteoric smoke particles is influenced by several processes, the influence of the atmospheric background wind being particularly important (Megner





et al., 2006). Model calculations show that due to atmospheric transport, the number density and size distribution of meteoric smoke particles vary with season in the polar regions (Megner et al., 2006, 2008; Bardeen et al., 2008). Their small size has made it difficult to observe them directly, and many observation techniques only manage to infer their existence. For example,

in-situ rocket measurements showed a depletion of the electron density in the main altitude ranges of meteoric smoke particles. It is assumed that charge neutrality is preserved as negative charges accumulate on the dust, and observed electron deficits are interpreted as an indication of dust particles (Friedrich et al., 2012). Charged dust particles are also measured by rocket-borne Faraday probes; the interpretation of these measurements is complicated, however, since the charge and fragmentation of the particles can also occur in the detector (Antonsen et al., 2017).

In the altitude range where these dust particles can be found, the ionospheric parameters are measured with radars by means of incoherent scatter. The incoherent scatter comes from the scattering of electrons that are coupled through charge oscillations to the other ionospheric components, including positive ions, negative ions, and charged dust; in addition, the collisions with the neutral atmosphere affect the incoherent scatter because they damp the charge oscillations, as the plasma is collisionally dominated. The role of charged dust particles in incoherent scatter has been studied so far only a handful of times. To describe

the incoherent scatter from the D-region ionosphere, an approach was developed that considers charged dust in addition to negative and positive ions (Cho et al., 1998b). This model approach has been used to derive estimates from incoherent scatter observations of dust size and positively charged dust number density (Rapp et al., 2007; Strelnikova et al., 2007; Fentzke et al., 2009). In recent work (Gunnarsdottir and Mann, 2021), we extended the description by Cho et al. (1998b) including a dust charge distribution; and investigated the influence of charged dust on incoherent scatter for the ionospheric conditions at the

EISCAT VHF radar site. We have found that conditions of high electron density in the winter months are best for studying the dust signatures in the spectrum. We also suggested supporting the analysis by using temperature information from independent measurements. This is due to the large influence that temperature has on the spectrum, of the same order as charged dust.

In this work, we present an analysis of incoherent scatter observations selected from the EISCAT VHF radar data archive to investigate the influence of charged dust on the spectrum and we attempt to derive a dust distribution. The paper is structured

as follows. In Section 2 we briefly describe the incoherent scatter model used and the radar data analysis approach. In Section 3 we describe the data used including radar observations, lidar observations, temperature and atmospheric models, the dust number densities obtained from a simulation run, and the dust charging model. Section 4 includes the data processing and analysis and Section 5 contains the conclusion.

## 2 Model of the incoherent scatter spectrum and selection of the observational data

If the number density of charged dust particles in the ionosphere is sufficiently large, they form dusty plasma and participate in incoherent scatter and influence the spectrum. Cho et al. (1998b) extended the incoherent scatter theory by Mathews (1978) to include charged dust. They developed an N-fluid description of the ionospheric plasma that includes a polydisperse charged dust component in addition to positive and negative ions. The shape of the radar spectrum depends on the electron density, mean ion mass, neutral density, dust size, dust charge distribution, and temperature of all constituents. The presence of positive



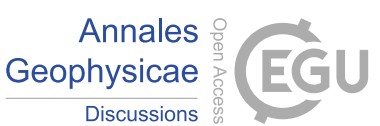

dust particles or large (>0.5 nm) negative particles causes the spectrum to narrow, while smaller (<0.5 nm) negative particles
    cause the spectrum to broaden. Dust only affects the spectrum if its charged population has a high enough number density
    compared to the electron density, so it changes the electron diffusion rate and consequently the spectrum (Cho et al., 1998b;
    Rapp et al., 2007). The plasma in the D-region is collisionally dominated, and so collisions with the neutral atmosphere affect
    incoherent scatter spectrum because they dampen charge oscillations. Furthermore, because the neutral density is high in the
D-region, the electrons and ions have temperatures approximately equal to those of the neutral gas.

    Gunnarsdottir and Mann (2021) extended the models developed by Cho et al. (1998b) to include dust with a charge dis-
    tribution. This approach was used and combined with a dust charge distribution model (Baumann et al., 2015) to calculate
    radar spectra and analyze the effect of charged dust on the spectra throughout the year. Comparison of the calculated spectra
    revealed that the influence of dust was most prominent in the winter spectra. Therefore, winter months in combination with a
high electron content in the ionosphere were the criteria for selecting the observation data.

    So far, the contribution of charged dust to incoherent radar scatter has been investigated only in a few cases. Most of these
    works investigated radar autocorrelation measurements. By fitting them with an adjusted Lorentz profile, a single dust size and
    the number density of positively charged dust were derived (Rapp et al., 2007; Strelnikova et al., 2007; Fentzke et al., 2009).
    Here, we consider the frequency spectrum, which is the Fourier transform of the autocorrelation function, and compare it
directly to the calculated spectra. We calculate the spectra using the neutral temperature as the temperature of all components;
    and obtain the neutral temperatures from available LIDAR observations or otherwise from a model. We used dust distribution
    data calculated with the WACCM-CARMA model (Bardeen et al., 2008), dust charging from model assumptions, and neutral
    densities from an empirical model of the upper atmosphere nrlmsise-00 (Hedin, 1991). To obtain the best fit, we vary the
    ion-neutral collision frequency and the amount of charged dust with the size distribution given by the WACCM-CARMA
model.

## 3 Observational and atmospheric model data

### 3.1 EISCAT VHF measurement

We chose an observation from 9 January 2014, 8-22 UT, with the EISCAT VHF radar (224 MHz) where a high amount of
particle precipitation is present; see figure 1. This large amount of particle precipitation could be connected to the strong solar
proton event on 6-9 January 2014 (NASA). Here, we see particle precipitation going below 80 km and enhanced electron
densities measured by the radar down to 65 km during the day. Two small dots, around 68 km (13:30 UT) and 78 km (21:30
UT), are not included in the data analysis, as they are unlikely to be from incoherent scatter.

### 3.2 LIDAR measurements of temperature.

In our previous study (Gunnarsdottir and Mann, 2021) we showed that the spectrum is highly influenced by atmospheric
temperature, so to accurately estimate the spectrum, we used the temperature measured by the Tromsø sodium LIDAR (Nozawa



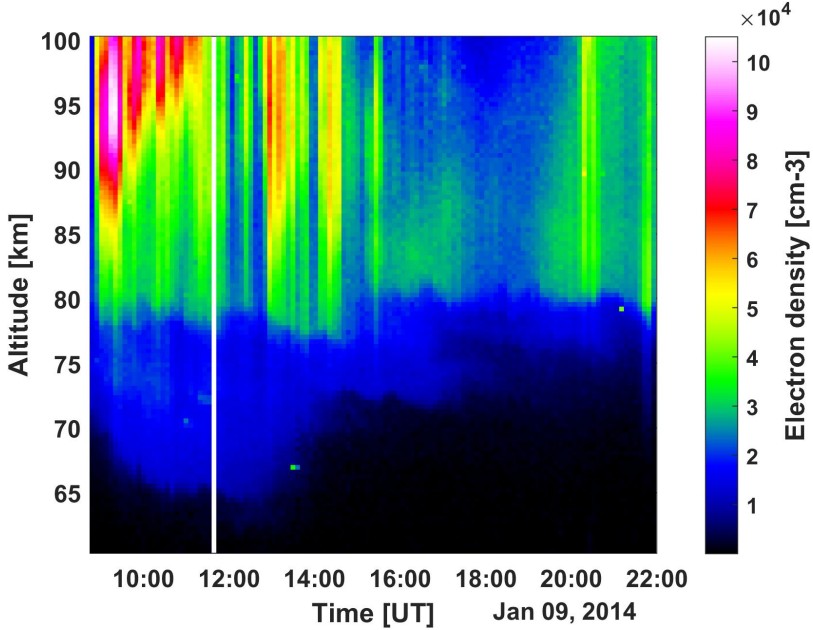

**Figure 1.** Electron density measured with the EISCAT VHF radar at 9 January 2014 from around 8-22 UT. White line in between 11-12 UT shows data removed due to some artifact in the measurement.

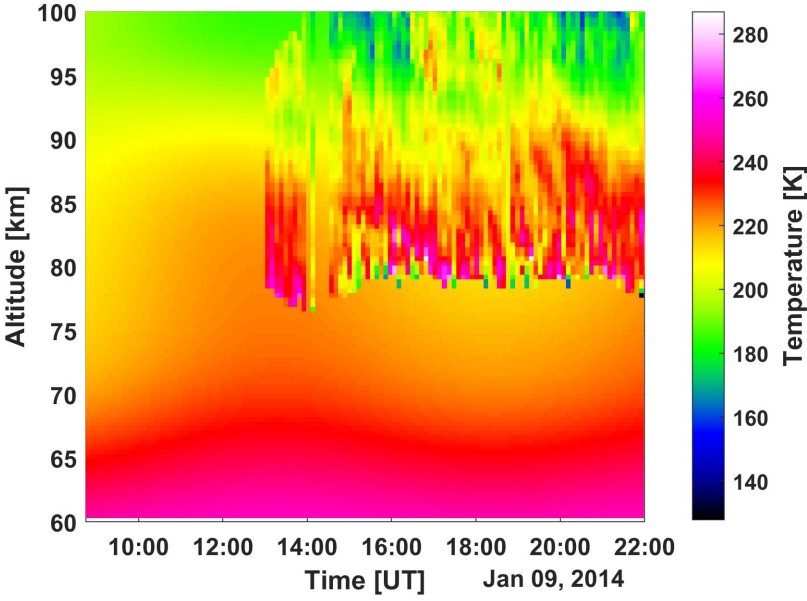

**Figure 2.** LIDAR temperature measured by the Tromsø Sodium Lidar on 9 Jan 2014 from 08-22 UT. Resolution is 6 min temporal and 0.5 km altitudinal. Only data points with error <5K are included in the plot and the data analysis. Where LIDAR temperature is not available we include model temperature from the nrlmsise-00 model.





et al., 2014). The LIDAR measured the temperature for only part of the observation time; thus we include the model temperature from the nrlmsise-00 model when there are no available LIDAR measurements. An overview of the temperature measured with the LIDAR and the added model temperature is given in figure 2. Here, we have only included LIDAR temperature measurements that have measurement error <5K. Temperature differences of, for example, 20-30 K can alter the spectrum in

a similar way as the charged dust does, and we therefore want to minimize the influence of the temperature. The comparison of the LIDAR measurements with the model temperature shows that this sometimes deviates and therefore all dust densities derived by using the model temperature have an additional uncertainty.

### 3.3    Dust density profiles from WACCM-CARMA

To fit the data with a charged dust profile, we start with number densities from a global atmospheric dust model (Brooke et al.,

2017; Plane et al., 2015; Hervig et al., 2017). The height profiles of meteoric smoke particles are derived from the Whole Atmosphere Community Climate Model (WACCM) (based on Hervig et al. (2017)) with a sectional microphysics model, the Community Aerosol and Radiation Model for Atmospheres (CARMA) (Bardeen et al., 2008). A meteoric smoke particle density of 2 $g/cm^3$ was assumed (Saunders and Plane, 2011). The model simulation was free-running for 21 years from 2000, enough time to reach a steady state of the model data. It used version 1 of the Community Earth System Model (CESM1) as a

common numerical framework (Hurrell et al., 2013). The model run is atmosphere-only simulations with interactive chemistry and aerosol forced with observed sea surface temperatures, etc. (Marsh et al., 2013). The simulation has a horizontal resolution of 1.9° (in latitude) × 2.5° (in longitude) on 66 $\sigma$-pressure vertical levels (1000-5.96×$10^{-6}$ hPa). The vertical resolution in the mesosphere and lower thermosphere is about 3.5 km. The resulting dust profiles have 28-size bins, with 0.2 nm being the smallest and 102.4 nm the largest, with monthly average dust densities. Figure 3 shows the dust densities for 14 size bins( 0.2

to 4.032 nm) for the altitude range 60-100 km in the first three panels. The last panel shows the total dust number density for the entire altitude range (including all sizes with number densities greater than 1).

Although the shown number density is the total average monthly number density for January, we do not know how much it is charged at any given time. Most rocket observations and model calculations suggest that dust particles are probably negatively charged (Rapp et al., 2012; Baumann et al., 2013). In the absence of direct observations, we use the charging probability based

on model calculations Antonsen (2019) and combine this with the dust number densities given by the WACCM-CARMA model. The charging probabilities are shown in Figure A2 in the Appendix. Based on these values, the smallest dust remains uncharged (<0.5 nm) and the resulting number density profiles of negatively charged particles in the 0.5-4 nm size range are shown in Figure 4. Here, the smallest particle sizes have a lower charging probability than the larger particles. Due to the large amount of particle precipitation seen in the observation, there might be additional charging processes occurring that,

without extensive modeling, we can only guess at. In a later section, we also discuss results obtained when using other charge distributions or charge polarity.





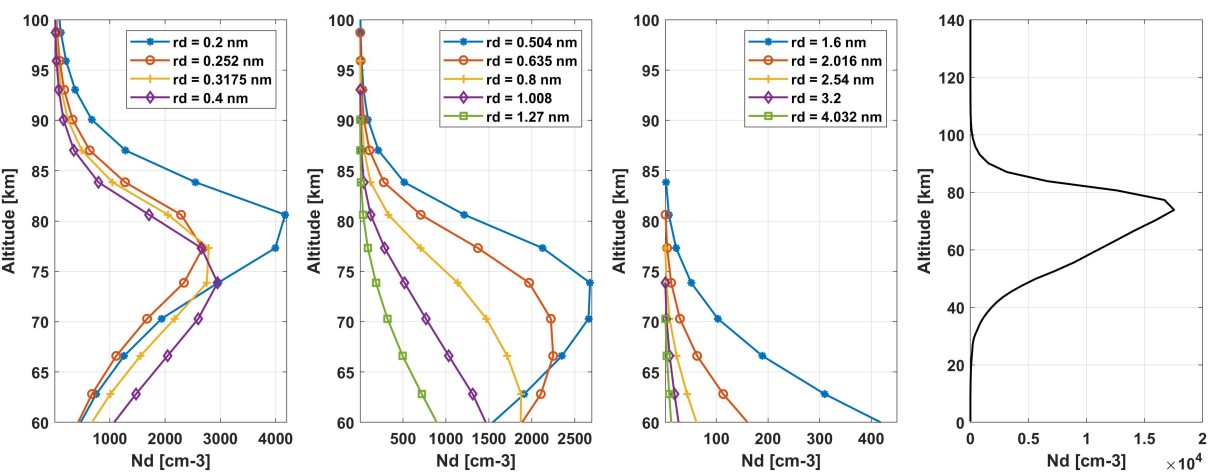

**Figure 3.** Dust distribution for sizes 0.2 - 4.032 nm in the altitude range 60-100 km from the WACCM-CARMA model shown in the first three panels. Last panels shows the total number density of dust for all sizes in the altitude range 0-140 km. Monthly average data for January with longitude and latitude closes to the EISCAT site.

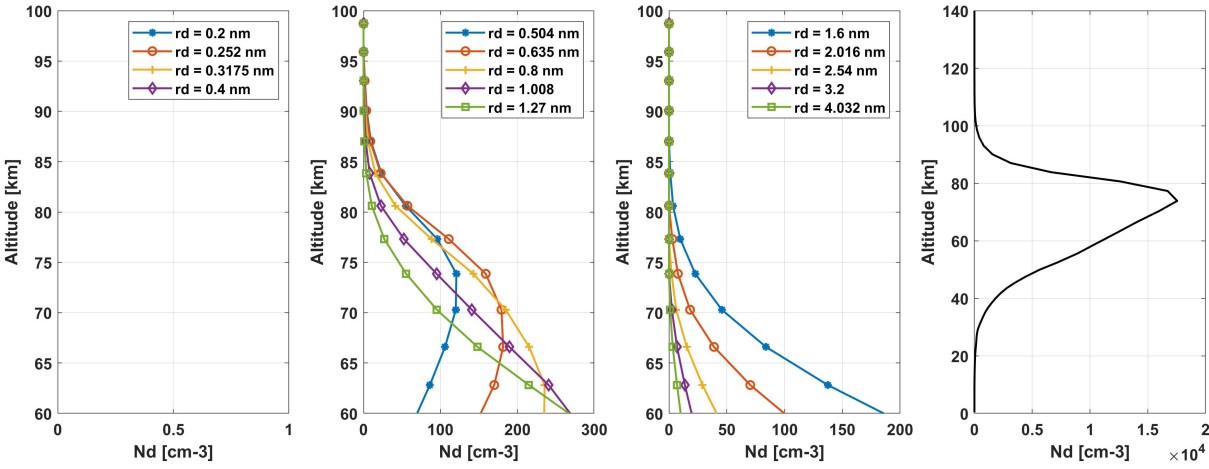

**Figure 4.** Estimated negatively charged dust distribution. We assume that dust particles below 0.5 nm have zero charge state and that sizes in the range 0.504-4.032 nm are charged according to the charge probability in Figure A2 for dust sizes without photodetachment.





## 4 Data processing and analysis

The dust signature in the radar data is quite small and difficult to detect, and many influences on the radar measurements can mask these dust signatures. Therefore, careful data processing is required to minimize noise that could distort the spectra.

Unwanted signals include echoes of meteors and satellites that pass through the radar beam. Using the Grand Unified Incoherent Scatter Design and Analysis Package (GUSIDAP) (Lehtinen and Huuskonen, 1996) we can improve the data by removing the presence of meteors in the raw signals. GUSIDAP has a built-in code that removes data influenced by satellites. This code can also be used to remove meteors by increasing the threshold of what is considered "bad data". The raw data are then run through the EISCAT Real Time Graph (EISCAT). to obtain the spectra. Here, we have chosen a time resolution of about 6.5 minutes and

the usual 360 m altitude resolution for the resulting spectra. Examples of spectra measured at two selected times of observation are shown in figure 5. Here, one can see some interesting features of the spectra. In the lower region, the spectrum appears to narrow at certain altitudes before widening with altitude, as expected. Above 90 km the spectrum becomes increasingly noisy due to large range resolution, and mostly below 75 km (after 14 UT) the electron density present is to low to discern a good radar signal. In the time interval 9-14 UT the increased electron density allows some spectra to be derived below 75 km. In

further data analysis, we remove cases that are entirely noisy, and smooth the cases that are heavily influenced by noise (using a Savitzky-Golay filter).

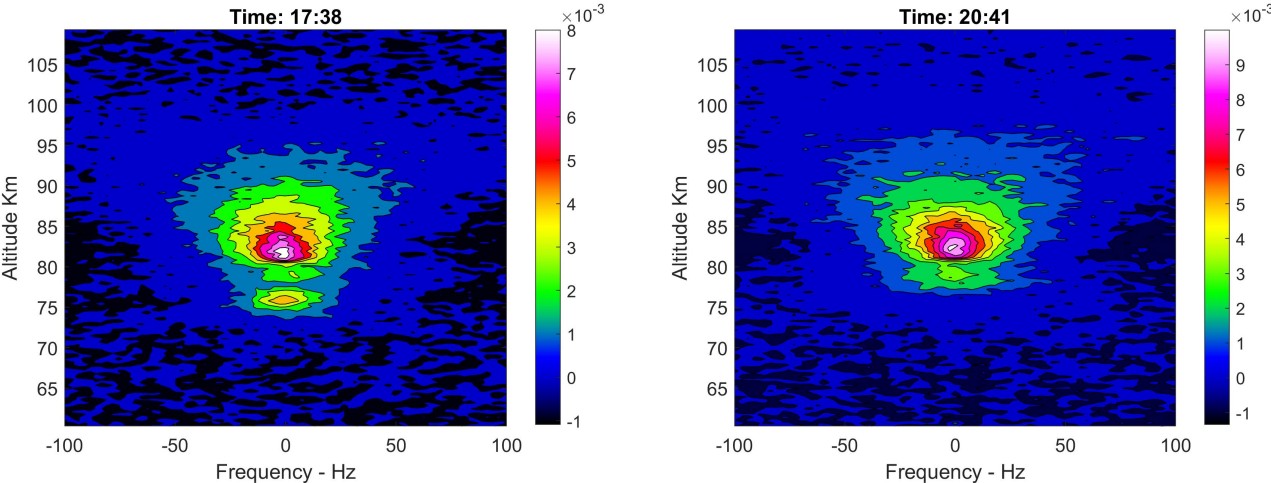

**Figure 5.** Measured spectrum by the EISCAT VHF for times 17:38 UT on the left and 20:41 UT on the right. The altitude range shown is 60-110 km. The spectrum is shown in a contour plot with arbitrary color-scales.

### 4.1 Modeled spectra without a dust component - adjusting the collision frequency

We first start by comparing the measured spectra with modeled spectra, where we assume that there is no dust component. An example of this is given in Figure 6, where we can see that the modeled spectra are too broad compared to the measured spectra





in the frequency range ± 10-30 Hz. The presence of large charged dust will narrow the spectrum by introducing a narrow peak on top of the normal broad background spectrum, and thus consequently decrease the spectral width. Here, we see, however, that many cases of measured spectra are actually narrower than predicted. However, they are narrower in the frequency range up to ± 50 Hz. And this cannot be fully explained by the presence of dust. It was, however, seen also in other observations. Recently, Thomas et al. (2023) noticed that the collision frequency modeled in the D region is off by a certain factor, which can

help explain this discrepancy between observation and model. Therefore, we run the model again with varying multipliers of the collision frequency (range used 0.1 to 3) and determine the best fit to the data in the frequency range ± 50 Hz. This results in the left panel of Figure 7, where the color scale represents the best-fit multiplier for the collision frequency with respect to the observed spectrum. The right panel shows the modeled ion-neutral collision frequency with this adjustment. Using the adjusted collision frequency when modeling the spectra, we get a better agreement with the measured spectra. This can be seen

in Figure 6 as the black line with circles, where for this particular case we have a very good agreement with the observation.

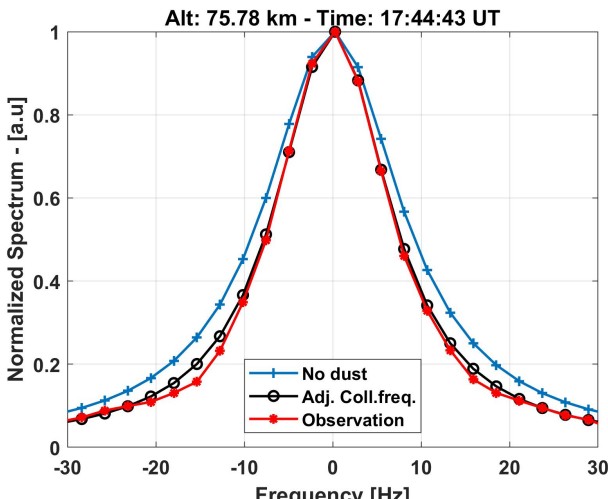

**Figure 6.** Comparison of a selected case of observed spectra (red circles) with a model calculation of a spectrum without a dust component (blue crosses) and the same model calculation with an adjusted collision frequency (black empty circles).

In Figure 8 we compare the estimated ion-neutral collision frequency (which depends on the neutral density and ion mass) using the neutral density from the nrlmsise-00 model, the adjusted collision frequency using the adjustment found above, and the collision frequency estimated from the IS spectrum fitting using GUISDAP. It is often difficult to derive the collision frequency from the IS spectrum fitting in the D- and E-regions, so we integrated the IS spectrum for 1 hour and derived it as

accurately as possible. The other two estimated collision frequencies were then averaged over 1 hour and compared with the GUISDAP results: the collision frequencies derived from the IS spectrum fit are sometimes an order of magnitude higher than the other two at 70-85 km altitudes (e.g. 10, 16 and 20 UT). However, due to the large IS fitting errors, the other two collision frequencies are also included in their error ranges.





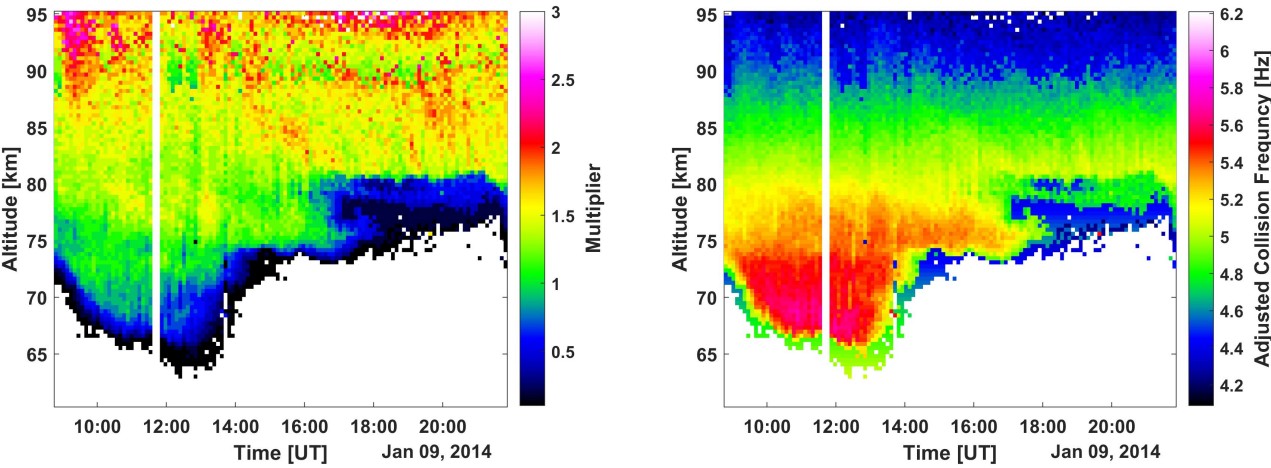

**Figure 7.** Estimated adjustment of the collision frequency where the left panels shows the factor needed to multiply the ion-neutral collision frequency to better fit the spectrum in the frequency range ± 50 Hz. And on the right is the adjusted model ion-neutral collision frequency.

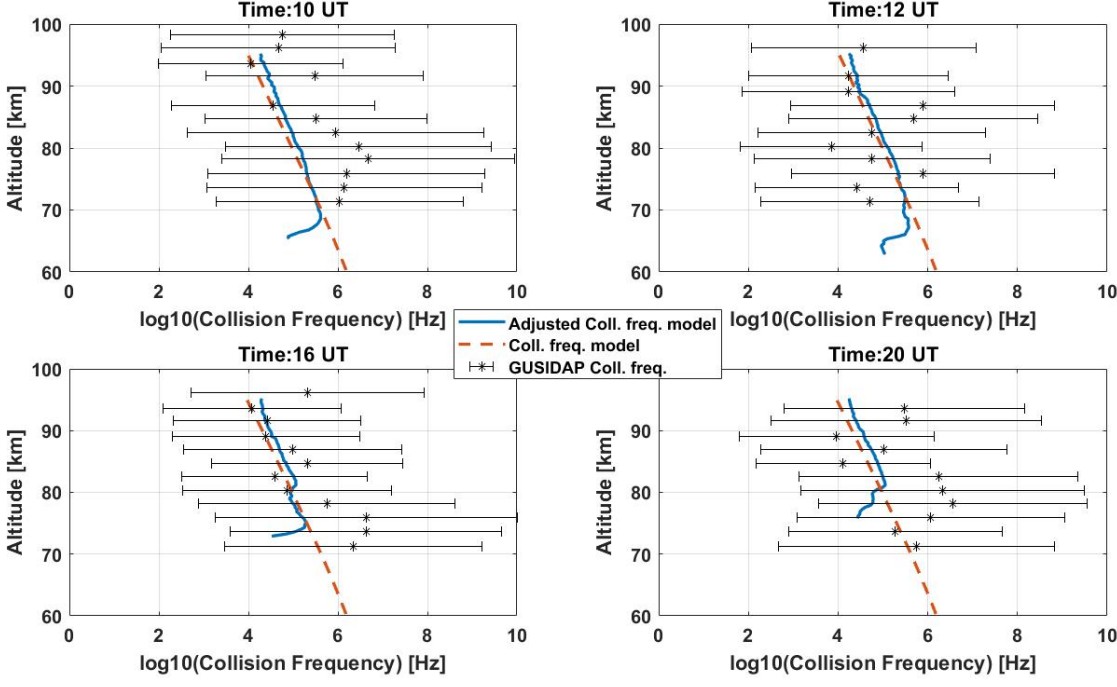

**Figure 8.** Comparison of the estimated model ion-neutral collision frequency (blue line), adjusted collision frequency (red dashed line) using the adjustment found above, and the collision frequency estimated from the IS spectrum fitting using GUISDAP (black stars with errorbars). See appendix Figure A3 for more detail on derived collision frequency.





## 4.2 Including dust in modeled spectrum

After adjusting the collision frequency, several observed spectra fit well with the modeled spectra in the frequency range ± 10-30 Hz, but often not in the most inner part of the spectrum, in the frequency range ± 10 Hz. This range is where we would expect a dust component to influence the spectrum (Cho et al., 1998a). We include a dust component in the modeled spectrum, with charged dust number densities shown in Figure 4 based on model assumptions outlined above to examine whether this will lead to a better fit in the frequency range ± 10 Hz. Due to the increased electron density seen during the observation,

we ran model calculations with several different variations of this number density. In this way, we could investigate whether a much larger amount of charged dust is required to fit the observed spectra or a smaller one. Figure 9 on the left shows an example spectrum where including a dust component leads to a better agreement with the observation. The charged dust number densities used to calculate the spectrum are shown on the left.

When including a charged dust component in the model, we assume that charge neutrality is kept. In practice, this means

that, since we assume the dust is negatively charged, we increase the ion population in the model to be equal to the sum of the number of charged dust components and the total electron density measured by the radar (derived by GUISDAP). We also assume a dust mass density of 2 $g/cm^-3$ (same as the WACCM-CARMA model). The ion mass is kept at 31 amu, which should be the mean ion mass above 80 km. Below 80 km the mean mass is assumed to vary for both negative and positive ions.

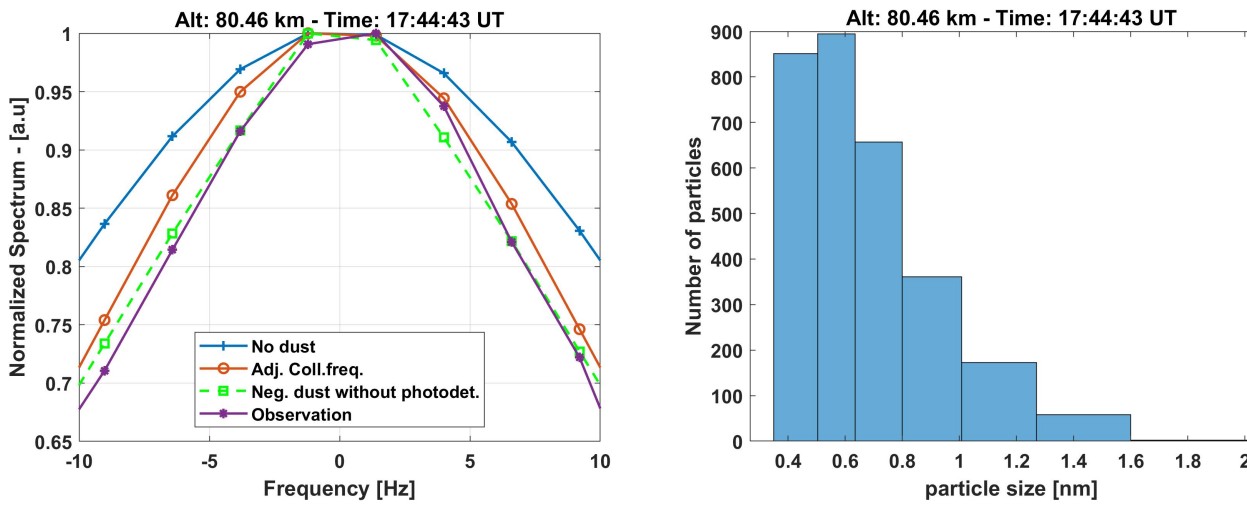

**Figure 9.** Comparison of a selected case of observed spectra (purple circles) with a model calculation of a spectrum without a dust component (blue crosses), the same model calculation with an adjusted collision frequency (red empty circles) and the a model calculation with and adjusted collision frequency as well as a charged dust component (green squares) is shown on the left. Time and altitude of the selected case is given above the figure. On the right is the associated size distribution ($cm^-3$) of the charged dust used in the model calculation on the left.

For each individual observed spectrum, we calculate spectra assuming different densities of negatively charged dust and

find the best fit to the observation. This is then compared to the model calculation of a spectrum without dust and to a model calculation with an adjusted collision frequency. In Figure 10 we show the cases where including dust in the modeled spectrum

results in the best fit of these three cases. The total number density (log scale) of dust assumed for the fit is given. As one can see, including dust in the model calculations fits better than the other two cases in quite a few cases in the altitude ranges of 75-85 km. After 17 UT the measured electron density is too low to obtain good measurements of the spectra at the lower

altitudes. A few cases are seen when the electron density is quite low below 75 km; here, however, the associated measurement error of the electron density is high and the number density of dust is low compared to the electron density, so that these fits are not very reliable (see the right panel of Figure A4 in the Appendix). The lack of knowledge of an exact mean ion mass could also introduce an additional error. The same can be said for fits above 90 km, where the range resolution is much poorer and the measured spectra are quite noisy.

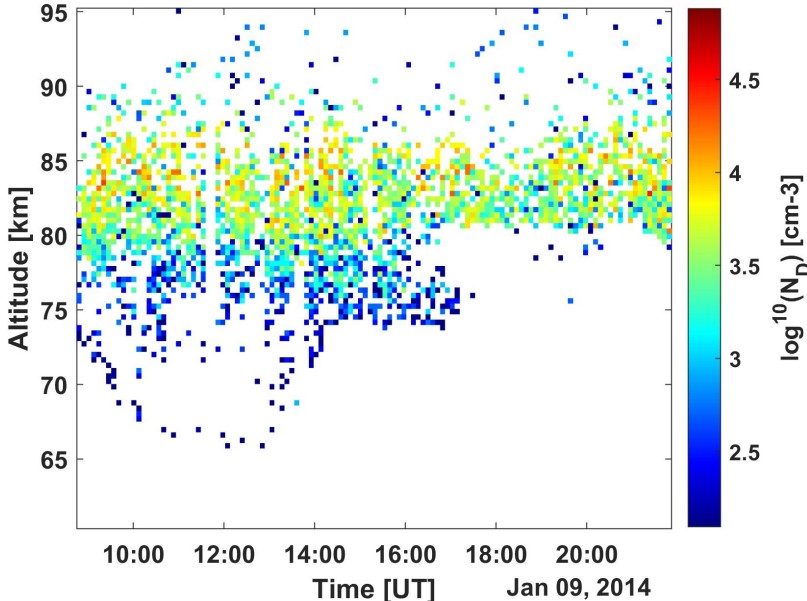

**Figure 10.** Derived number density ($cm^-3$) of negatively charged dust needed to fit to measured spectrum, shown for the time and altitude range of the observation. The quality (residuals) of these fits are shown in Figures A6-A12 in the Appendix.

Using data from Figure 10 we show the average dust number density derived for charged dust and compare it with the total dust number density from the WACCM-CARMA model (Figure 3 and the average measured electron density. Here, we can see that the average dust density needed follows the total modeled number density above 85 km, but below this altitude the number density decreases, as does the average electron density. The lowest number densities of dust at high and low altitudes are unreliable, as has been discussed. It seems that a peak of the average number density occurs around 85 km. We

also considered the influence of the assumed temperature on the result. Comparison of the number densities using the LIDAR temperature and the model temperature (see Figure A5) shows that in the main altitude range where we see dust particles, the dust number density needed is lower for the cases modeled with the LIDAR temperature. This is due to the use of higher temperature measurements (the LIDAR temperature at that altitude is slightly higher than the model temperature), where the higher temperature causes a broader spectrum, and thus the number densities using the model temperature can be too high. In




the right panel of Figure 11 the average dust size is shown, with an increasing average dust size with decreasing altitude. Other

methods for determining dust size from radar measurements have shown dust radii close to 1 nm throughout the altitude range (Strelnikova et al., 2007; Rapp et al., 2007).

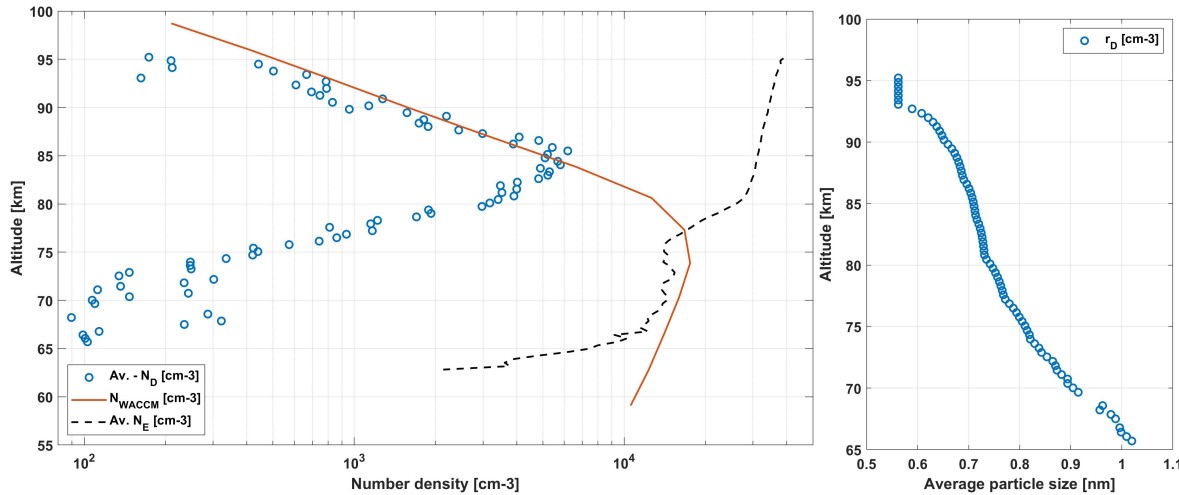

**Figure 11.** Derived average dust number density (Figure 10) for each altitude compared to the total WACCM-CARMA dust number density (Figure 3) and the average measured electron number density (Figure 1) shown in the left panel. The right panel shows the derived average dust sizes corresponding to the average dust densities in the left panel.

Here, we have assumed that the dust is charged negatively in the range of 0.5-4 nm. However, the model relies heavily on charge neutrality, and thus, by including a positive dust component instead, we get similar average sizes, but with a slightly

reduced number density. The average positive number density has a shape similar to the negative average number density distribution in Figure 11. This is due to the fact that when a positive dust component is included, the ion population is reduced in the model and the entire range of positive dust sizes will narrow the spectrum, while small negative dust will broaden the spectrum. Large positive and negative particles influence the model in the same way for each size. It is mainly the decrease of the ion component when positive dust is included, that a reduced number density of positive particles is needed compared to a

negative component. Therefore, we cannot say whether any dust present is positively or negatively charged or a combination of both. Only that, if the dust is positively charged, the number densities would be lower than estimated here. According to the Baumann et al. (2015) model, the dust appeared to be mostly positively charged below 80 km during the day and negative particles were found at higher altitudes with a higher number density at night. However, their results were during relatively quiet ionospheric conditions in September, as our observation is in January with apparent high amounts of particle precipitation.

Therefore, we cannot conclude where and how much positive and/or negative dust might reside.

We have assumed that the small particles remain uncharged and consequently do not influence the spectrum. If the smallest dust particles were charged negative, they would cause a broadening of the spectrum, and we would need a larger number density of large charged dust to narrow the spectrum adequately to fit the observation. This additional broadening would be





difficult to distinguish from an additional larger dust distribution. Including small positive dust particles would narrow the
spectrum, however, due to the size dependence of the model, a very large number of small charged population are needed to
narrow the spectrum enough to fit the observed spectrum.

## 5   Conclusions

Our analysis strongly suggests that the incoherent scatter spectra in the considered height interval are influenced by the presence
of charged dust particles and their amount is of similar order as suggested by models. For the spectra that we calculated to fit
the observations, we assumed a dust component calculated with the WACCM-CARMA model with dust height profiles from
60 to 100 km and 28 size bins. We assumed a charging probability that varied with size and zero charges for particles smaller
than 0.5 nm and varied the absolute dust number density by multiplying the WACCM-CARMA profiles by a constant factor.

We could best evaluate the observations at heights 75 to 85 km. Only a fraction of the observed spectra could be analyzed
for higher altitudes, where the observations are limited by the low-altitude resolution of the data used, and at lower altitudes,
where the observations are limited because of low electron densities.

We have analyzed the incoherent scatter spectra observed with the EISCAT VHF radar in a selected time interval during
ionospheric conditions with a high electron content in winter. The winter season was chosen because when applying model
assumptions on the annual variation of the dust, its size distribution and height profile in winter are favorable for generating
clear signatures in the spectra. The observation was made from 9 January 2014 approximately 8:00 to 22 UT, after several days
with high solar flare activity, which we assume caused the unusually high electron content low down in the atmosphere.

Considerable electron densities were observed for some of the observation intervals even at altitudes as low as 65 km.
We investigated the obtained individual spectra in the range of the ion line and after meteor and satellite subtraction and
collision frequency correction fitted them with a spectrum including a charged dust component. The temperatures entered
in the calculations were taken from LIDAR observations made at the same location as the radar and the temperature of the
nrlmsise-00 model when no LIDAR measurements were available. where there is an indication of a lower dust number density
using the LIDAR data in the altitude area 75-85 km.

When investigating individual spectra, we found that a large fraction of them were too narrow compared to calculated spectra
over a rather large frequency range ($\pm$ 50 Hz). This could not be explained solely by the influence of a charged dust component.
The spectra were better reproduced when the ion-neutral collision frequency assumed for the model calculations was varied
with factors roughly 0.5 up to 2. Running for comparison a GUISDAP analysis with the collision frequency as a free parameter
led to similar results. This mismatch of the collision frequencies was also observed in D-region studies carried out by other
groups (Thomas et al., 2023). A possible explanation could be that the applied incoherent scatter models do not sufficiently
describe the collisions of the different ionospheric constituents, which are paramount at these altitudes because of the high
neutral density.

When including negatively charged dust particles in size ranges of 0.5-4 nm, we see a possible dust layer in the altitude range
of 75-85 km with a few good fits below 80 km when the electron density is high enough to produce a good enough spectrum.





A comparison to modeled spectra without dust shows that assuming a dust component improves the fit in the frequency range $\pm$ 10 Hz around the peak of the spectrum.

In view of possible future investigations, we note that the neutral density and temperature are best measured independently with other instruments to ensure a good analysis of the spectra. Temperature is quite variable on short timescales due to atmospheric dynamics. Therefore, the combination of radar and LIDAR studies would be helpful. Furthermore, the derivation of the total dust distribution is based on assumptions about dust charging. Especially in the observations studied here, the ionospheric conditions are far from typical, which leads to further uncertainties regarding the charge, which is based on model assumptions anyway. The derived number density and the average size also depend on the assumed dust input parameters. The average dust size is highly dependent on all the small negative dust particles included. Due to their small size, they do not influence the spectrum as much as the large particles and mainly influence its amplitude, while the larger particles narrow the spectrum (Gunnarsdottir and Mann, 2021).

The present study was carried out with data taken with the Manda radar code (Tjulin, 2017). The Manda code is well suited for studying layers in the mesosphere, but measures the ionosphere higher up with low resolution. Different radar codes should be considered for future studies. Since the EISCAT_3D radar measures at similar frequency, our study can be used to estimate the conditions for this new instrument. The transmit power of the phase of the new radar is about a factor of 3 higher than that of the system used in this study, so that the quality of measured spectra may improve. Our study shows, in line with other recent investigations, that the incoherent scatter from the D-region is not sufficiently described with the assumptions on collision rates in the present models used for analysis. Here theoretical investigations could be helpful as the D-region spectra are difficult to understand and influenced by several different parameters, all of which are variable and partly interrelated.

*Code and data availability.* EISCAT data are available under https://madrigal.eiscat.se/madrigal (accessed on 14 July 2023).

*Author contributions.* TG outlined the work, selected the observation time, and wrote the original draft of the paper. TG and IM conceived the idea for this work. TG, DH, EH, SN, YO and IM discussed the EISCAT observations and analyses. YO carried out GUIDAP analysis to derive collision rates. NS, SN, and TK provided lidar measurements used. WF provided WAACM-CARMA data used. All authors have read and agreed to the published version of the manuscript.

*Competing interests.* At least one author is serving on the editorial board of the journal. The authors declare that there is no conflict of interest otherwise. The funders had no role in the design of the study; in the collection, analysis, or interpretation of data; in the writing of the manuscript, or in the decision to publish the results.



*Acknowledgements.* This work was carried out within a project funded by the Research Council of Norway, NFR 275503. Norwegian

270 participation in EISCAT and EISCAT3D is funded by the Research Council of Norway, through the Research Infrastructure Grant 245683. The EISCAT International Association is supported by research organizations in Norway (NFR), Sweden (VR), Finland (SA), Japan (NIPR and STEL), China (CRIPR), and the United Kingdom (NERC). DH was funded during this study through a UiT The Arctic University of Norway contribution to the EISCAT3D project. The sodium LIDAR at Tromsoe (Nozawa et al., 2014) has been operated under collaborations of Nagoya University, RIKEN, Shinshu University, and The University of Electro-Communications. NS, SN, TK have been supported by a

275 Grant-in-Aid for Scientific Research A (21H04516), B(17H02968). NS, SN have been supported by a Grant-in-Aid for Scientific Research B (23H03532, 21H01144). SN has been supported by a Grant-in-Aid for Scientific Research A (21H04518), and a Grant-in-Aid for Exploratory Research (20K20940). WF was supported by the NERC project NE/W003325/1. The WACCM/CARMA model simulation was performed on the Leeds ARC HPC facilities.

## Appendix A:  Appendix figures

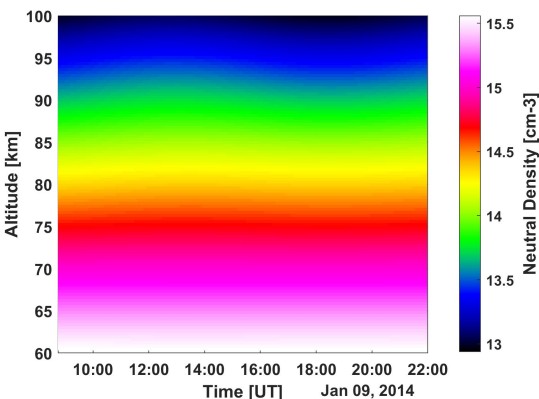

**Figure A1.** Neutral density used in the model calculations of the spectrum and ion-neutral collision frequency. From the nrlmsise-00 model, using F107 = 188.2, F107 monthly = 151.2, APH = 8

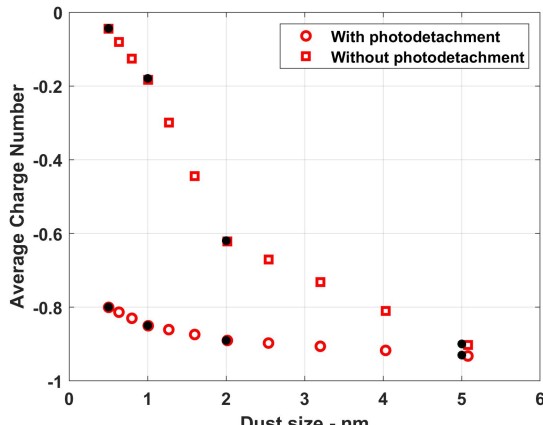

**Figure A2.** Average charge number for negative dust particles of sizes 0.5-5 nm. The cases shown are with and without photodetachment. The black dots / squares are based on the charging model of Antonsen (2019) and the red data is interpolated to match the size bins from the WACCM-CARMA model.



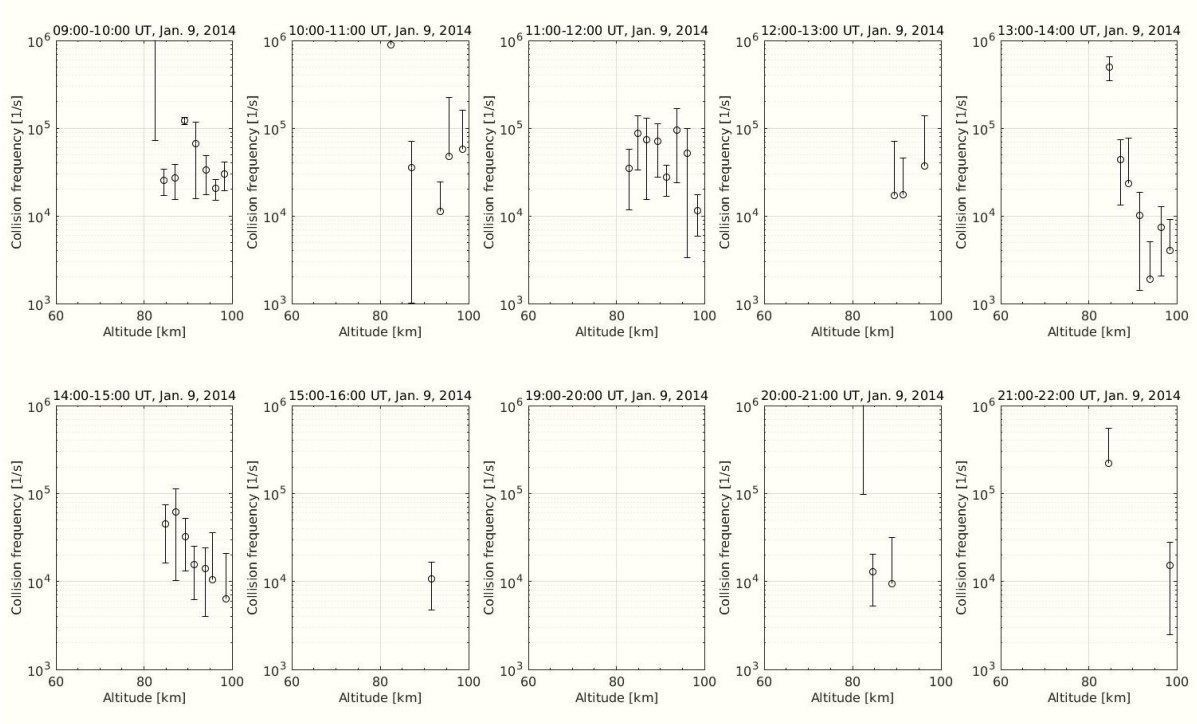

**Figure A3.** collision frequency estimated from the IS spectrum fitting using GUISDAP. The IS spectrum is integrated for 1 hour and derived as accurately as possible.

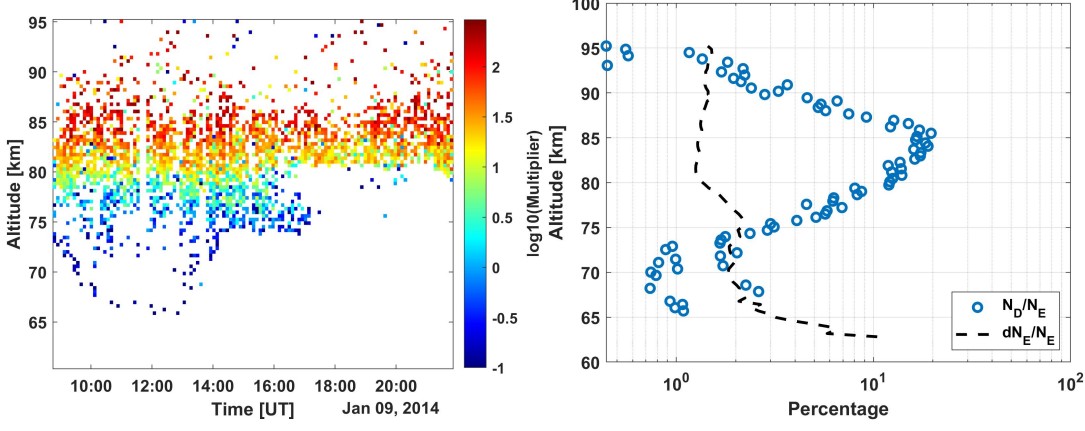

**Figure A4.** The left panel shows the charge multiplier needed to fit the spectrum of cases shown (log scale). Used for figure 10. The right panel shows the relative number of charged dust to electron density ( average) and the relative error of the derived electron density with the EISCAT VHF radar (See Figure 11).





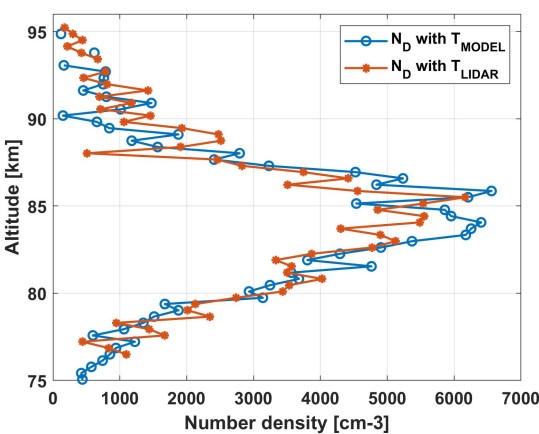

**Figure A5.** Average derived dust number densities (cm-3) using model temperature from nrlmsise-00 model and the LIDAR temperature respectively.

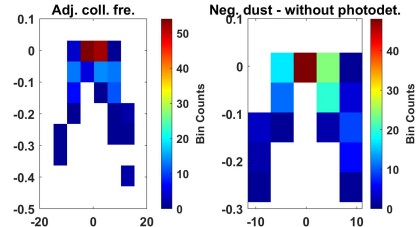

**Figure A6.** Residuals for 60-65 km. Model points minus observational points around the zero frequency (11 measurement points). Only showing best fits for this altitude segment.

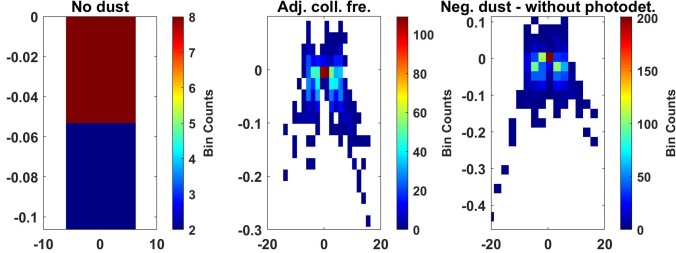

**Figure A7.** Residuals for 65-70 km. Model points minus observational points around the zero frequency (11 measurement points).





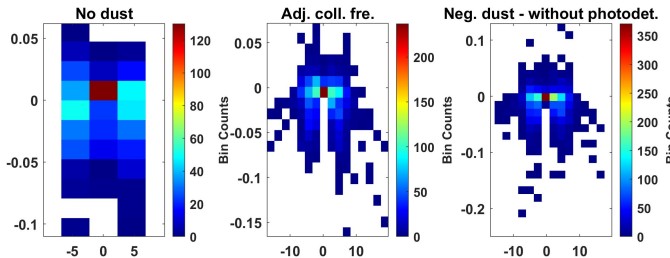

**Figure A8.** Residuals for 70-75 km. Model points minus observational points around the zero frequency (11 measurement points).

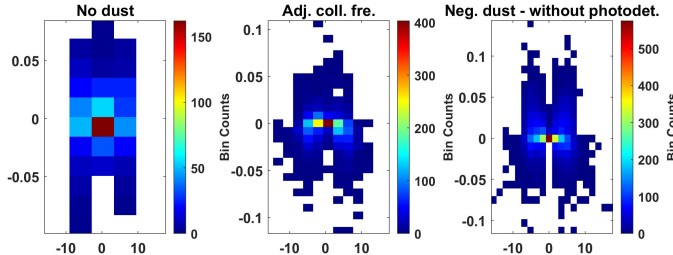

**Figure A9.** Residuals for 75-80 km. Model points minus observational points around the zero frequency (11 measurement points).

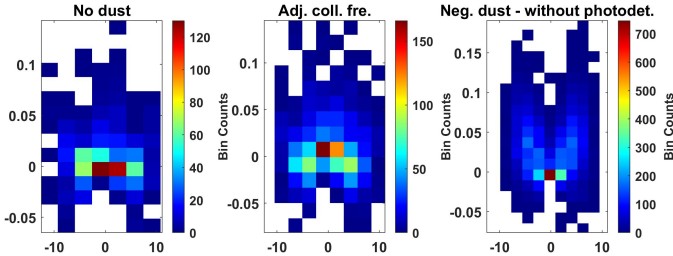

**Figure A10.** Residuals for 80-85 km. Model points minus observational points around the zero frequency (11 measurement points).

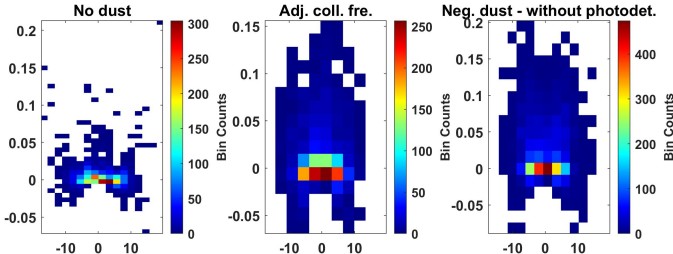

**Figure A11.** Residuals for 85-90 km. Model points minus observational points around the zero frequency (11 measurement points).

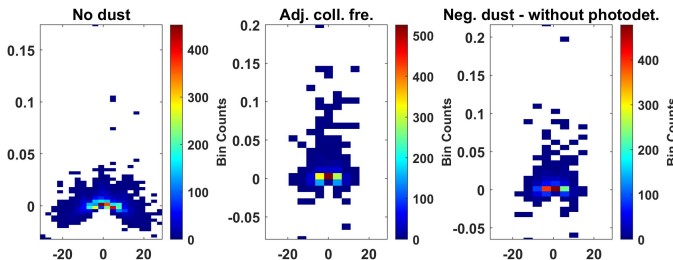

**Figure A12.** Residuals for 90-95 km. Model points minus observational points around the zero frequency (11 measurement points).

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
