# Peer review of "Influence of Meteoric Smoke Particles on the Incoherent Scatter Measured with EISCAT VHF"

_Annales Geophysicae, 2023_

## Author Response (AR1)

We thank the reviewers for the constructive comments and list our response below together with the specific comments:

**Line 64: Can you give a short physical explanation of why the influence of dust was most prominent in the winter spectra ?**

The influence of the dust on the spectrum depends on the heigh profile of dust particles. When assuming the height profiles predicted by models then the dust influence on the spectrum is more evident. We showed this in a previous work and will describe this more clearly when revising the manuscript.

**Line 77, Section 3.1: Here I would expect to see more details of the radar parameters. For example was the antenna pointed vertically ? What type of radar coding was used ?  What are some of the details like the range and frequency resolutions etc. The text near the end of the paper (lines 253-254) mentions some of thes things and  should be repeated and expanded  upon in this section so that readers¨ can better understand details of the data shown.**

The radar code was MANDA and the radar pointed vertically, we will modify the text accordingly.

**lines 98-99: The previous sentence states that a height profile of smoke particles is used from a model.The assumed 2 g/cm3 smoke density must then be specified at some height, I presume. Or is this a total density that is uniform thoughout the whole height region ? Please elaborate.**

The model output describes numbers per volume of particles in different size bins and as function of height. The given density is the bulk material density that we used to convert from mass to size of particles assuming they have spherical shape. This is needed because size and mass enter the description of the model. We will make this clear in the modified text.

**Fig.4, caption: The last panel is clearly not a negatively charged dust distribution, but the same dust distribution as that in Fig. 3. This needs clarification in the caption, or the figure should be changed. Perhaps the last panel needs changing to show the total charged dust distribution.**

Indeed, the last channel describes the total dust density (same as shown in Fig. 3) that we repeated here for illustration. The last figure has been replaced with the total number density of the assumed charged dust.

**Lines 127, 254: I don't think large range (altitude) resolution is the reason for poor spectra above 90 km**

**According the the description of the manda modulation given in Tjulin (2018, or 2022 for the latest corrected version) (https://eiscat.se/wp-content/uploads/2022/02/Experiments_v20220203.pdf) the basic range resolution is 0.36 km everywhere, (although in the analysis ranges may be averaged together). The main reason for the poor spectra is probably that the incoherent scatter spectrum becomes too wide as one increases in altitude for for the spectral range of the data channels analysed here (+/- 330 Hz).**

**The other, perhaps more natural reason, is that the backscatter simply is weaker in this height range, at least for the times chosen in Fig. 5. That the signal is strong below 90 km appears to be caused by the auroral precipitation.**

**Please check this, with your co-author IH for example.**

We agree with this comment. We checked the observed spectra for two different spectral ranges (wide and narrow) and can confirm that the width of the spectrum often leads to poor spectra at high altitudes. We also confirmed that the range resolution is fixed to 0.36 km for the data considered in this work. We will modify the text accordingly.

**Fig. 6: Was the adjusted collision frequency a simple multiple of the model frequency ?**

**If so it would be informative to specify what value the multiplier had.**

This will be corrected. We will include value of the multiplier used for the figure (1.2).

**Fig. 8 caption: "(blue line)" and "(red dashed line)" are in the wrong order. The figure legend is correct.**

This is corrected.

**Line 163: Should it not be ".. are shown on the right" ?**

**Section 4.2 (from line 169): If I understand correctly, three spectra are calculated: 1) with dust and model collision frequency 2) without dust and model collision frequency and 3) without dust and adjusted collision**

**frequency. The case of dust with adjusted collision frequency is surely another possibility, but I suppose there are too many unknown variables for this to work. Could you expand on some of these details here: e.g. in roughly what fraction of cases are the dust spectra the best fit (you write "quite a few cases") compared to the other two cases ? It would also be intereting to know in what fraction of cases did the adjusted collision frequency model and the model collision frequncy give the best fit.**

Our overall finding is that adjusting the collision model improves the agreements at the flanks of the spectrum while the dust component improves the agreement between model and observation in the center of the spectrum. We will discuss this more clearly in the revised manuscript.

**Fig. A4: This is the first reference to a "charge multiplier" needed to fit the spectra. A multiplier for the collision frequency was used earlier.**

**Please explain what this charge multiplier does in the main body of the paper. Does it multiply the average charge number without photo-detachment in**

We will include a discussion on the charge assumptions in the revised version of the paper.

**Fig. A2 ? Does the fact that the multiplier goes negative in the lower part of the plot (below 75 km) imply that these are positive**

**dust particles ?**

This is rather due to the uncertainty of the spectra, and we show the result just for completeness.

**Figs. A6-A12: Please state what is actually plotted here. The x axis is obviously frequency (but please label the axes or state in caption).**

This has been corrected in the manuscript now with a more detailed figure captions.

**The Y axis shows the residual. Are the 11 measurement points 11 spectral values ? If so, why do the various plots show different numbers of frequency values**

**ranging from 1 (Fig. A7, left plot) to about 22 (Fig. A12, left plot) ?**

They show different number of values since, for each altitude range chosen, only the "best-fit" spectra are used for the residual plots. We can make this more clear in the text.

**Technical Corrections**

Thank you, we will correct these.

=====================

lines 36, 157 and 297: The Cho et al. 1998b reference is identical to the Cho et al. 1998a reference.

Should it be just one reference to Cho et al. 1998 ?

Fig.1 caption: Insert "The" before "white"

Fig.2 caption: I presume the smooth background colour is the model and the more variable

data segments are the Lidar data. This should be stated in the caption.

Fig.3 caption: "Last panel shows..." and "closest to..."

Line 124: Remove the full stop after "(EISCAT)"

Fig. 5 caption: Include the date of the measurements.

Line 153: I suggest re-wording to ".. are within their error ranges"

Line 181: The brackets need closing, presumably after "Figure 3"

Should there not be a reference to Fig. 11 where it says "Here..." ?

Line 263: Spelling of "GUISDAP"

Line 349: The Tjulin refence could be updated to 2022:  https://eiscat.se/wp-content/uploads/2022/02/Experiments_v20220203.pdf

Appendix: Fig. A1 is not referred to anywhere in the text.

All the above comments have been andressed in the final revision of the paper.